# Cell Migration–Proliferation Dichotomy in Cancer: Biological Fact or Experimental Artefact?

**DOI:** 10.3390/biology13100753

**Published:** 2024-09-24

**Authors:** Abdulaziz Alfahed

**Affiliations:** Department of Medical Laboratory, College of Applied Medical Sciences, Prince Sattam bin Abdulaziz University, Alkharj 11942, Saudi Arabia; a.alfahed@psau.edu.sa

**Keywords:** migration–proliferation dichotomy, proliferation index, migration index, gene set enrichment analysis, gastric cancer, colorectal cancer

## Abstract

**Simple Summary:**

Migration–proliferation dichotomy (MPD) is a characteristic that has been observed in cancer cells growing in artificial culture media. The MPD principle states that when cells begin to grow in number (cell proliferation), their ability to migrate from one point to another (cell migration) is paused, and vice versa. This phenomenon has been used to study cancer advancement in cancer cells growing in culture, and is proposed to be responsible for the adverse therapeutic outcomes observed in patients with cancer. However, MPD has not been comprehensively investigated in cancer tissues obtained directly from patients. This study investigated MPD in stomach and bowel cancers using gene expression signatures of natural cancers. The overall findings confirm that cell proliferation and migration occur in the reverse direction in cancer cells, which is in keeping with the features that have long been observed in artificially grown cancer cells. These results also provide a basis for further investigation of MPD in natural cancers.

**Abstract:**

The migration–proliferation dichotomy (MPD) has long been observed in cultured cancer cells. This phenomenon is not only relevant to tumour progression but may also have therapeutic significance in clinical cancer. However, MPD has rarely been investigated in primary cancer. This study aimed to either confirm or disprove the existence of MPD in primary cancer. Using primary gastric, colorectal and prostate cancer (GC, CRC and PCa) cohorts from the Cancer Genome Atlas and Memorial Sloan Kettering Cancer Center, this study interrogated the MPD phenomenon by utilising RNA–Seq-based proliferation (CIN70 signature) and migration (epithelial-mesenchymal transition) indices, as well as gene set enrichment analyses (GSEA). Alternative hypothetical migration–proliferation models—The simultaneous migration–proliferation (SMP) and phenotype–refractory (PR) models—were compared to the MPD model by probing the migration–proliferation relationships within cancer stages and between early- and late-stage diseases using chi-square and independent T tests, z-score statistics and GSEA. The results revealed an inverse relationship between migration and proliferation signatures overall in the GC, CRC and PCa cohorts, as well as in early- and late-stage diseases. Additionally, a shift in proliferation- to migration dominance was observed from early- to late-stage diseases in the GC and CRC cohorts but not in the PCa cohorts, which showed enhanced proliferation dominance in metastatic tumours compared to primary cancers. The above features exhibited by the cancer cohorts are in keeping with the MPD model of the migration–proliferation relationship at the cellular level and exclude the SMP and PR migration–proliferation models.

## 1. Introduction

Of the eight cancer hallmarks described to date [1], proliferation and invasion/migration appear to be the two most obvious ones, at least from the standpoint of cell cultures [2,3,4,5,6,7,8,9,10,11]. However, it has long been observed that cancer cells in culture exhibit a migration–proliferation dichotomy (MPD), in which the phenotypes of migration and proliferation in individual cancer cells have a mutually exclusive relationship [3,4,5]. The implication of this phenomenon is that for individual cancer cells, there is a decoupling of migration and proliferation, such that a cell pauses proliferation when it assumes the migration phenotype and vice versa. Although both proliferation and migration share common signalling pathways [3], certain factors have been implicated in the switch from migratory to proliferative phenotypes, including the Arg/Abl2 non-receptor tyrosine kinase [6], CSF/CSFR1 signalling [7] and microenvironmental cues such as hypoxia and inflammation [3,8,9]. Currently, the switch mechanism favoured by proponents of the MPD phenomenon is a transitional plasticity or transcriptional (epigenetic) switch rather than a mutation-driven change [3]. The switch mechanisms have been identified as epithelial-mesenchymal transition (EMT) [10] and reverse mesenchymal-epithelial transition (MET) [11]. However, the existence of mesenchymal molecular subtypes of malignant epithelial tumours, such as gastric and colorectal cancers (GC and CRC), in which primary tumour cells have mesenchymal and invasive/migratory phenotypes de novo, may be evidence of the MPD phenomenon [12,13].

It is instructive to note that most of the evidence for the MPD phenomenon comes from cell culture and tumour xenograft studies. However, cell line studies are hampered by certain limitations, such as the introduction of additional genetic and epigenetic changes in long-term cultures, the limitation of sample size to two or more cell lines per study, and the fact that cell propagation in culture may favour one cell type or state over the other [14,15]. These limitations of cell culture studies make it impossible to capture the full spectrum of tumour biology. In contrast, investigating the molecular mechanisms of cell migration and proliferation in clinical cancer is fraught with technical limitations and restricted to the use of a few immunohistochemical markers [16]; hence, there is a need to validate the MPD phenomenon using alternative tools that more robustly interrogate it in real-life primary cancer cohorts. High-throughput gene expression profiling, such as RNA-Seq, offers the opportunity to comprehensively examine the MPD phenomenon in a large cohort of primary cancers [17,18] and circumvent the technical limitations encountered with immunohistochemistry methodology and cell culture studies.

This study aimed to answer the question of whether cell proliferation and migration are mutually exclusive events in individual cells in the body of primary cancer, using gene expression signatures for cell proliferation and migration/motility. The study hypotheses are that the cell proliferation and migration programmes in individual cancer cells are mutually exclusive to each other and that this mutual exclusivity is reflected as an inverse, reverse or indirect correlation between proliferation and migration signatures at the cell population level in primary cancers. The results confirm the inverse relationship between migration and proliferation indices and support the MPD phenomenon observed in cell culture studies.

## 2. Materials and Methods

### 2.1. Principles

The possible hypothetical alternative migration–proliferation models for the MPD model are (i) simultaneous migration and proliferation at the individual cancer cell level (the SMP model) [19,20] and (ii) the two-separate-tumour-cell-clones model, in which one tumour cell clone is migration-proficient and proliferation-refractory, while the other clone has the reverse phenotype (the phenotype–refractory [PR] model).The implication of the MPD model in cancer cells is that the proliferation and migration phenotypes would either exist in a mutually exclusive relationship at the individual cancer cell level [3,4,5] or in an inverse relationship if measured by gene expression indices in a tumour body that also includes stromal and inflammatory cells. The SMP model implies that proliferation and migration phenotypes exist in direct correlation since individual cells can migrate and proliferate simultaneously [19,20]. In the PR model, proliferation and migration are independent of each other since proliferative cells are migration-refractory or migration-deficient, and vice versa. However, cell proliferation promotes the migratory phenotype because the higher the proliferation of cancer cells, the worse the depletion of nutrients and oxygen in the tumour microenvironment, and hence, the stronger the cues for migration [21,22].The body of a tumour in progression comprises subpopulations of cancer cells that either differentially or simultaneously exhibit proliferative and migratory traits at the cellular level. The dominant characteristic of a tumour body, i.e., proliferation versus migration/invasion at the cell population level, is dependent on the proportion of cancer cells with either cancer trait. However, this dominance would be absent if tumour cell proliferation and invasion are coupled together or occur simultaneously at the level of individual cancer cells.Mutual exclusivity between proliferation and migration per cell implies that an inverse relationship exists between proliferation and invasion/migration gene expression indices at the cell population level, both within each cancer stage and between early and late cancer stages. It also implies that differential enrichment of the proliferation and invasion gene sets must also exist between early- and late-stage diseases, inasmuch as early-stage disease is predominantly proliferative, whereas late-stage disease may or may not be largely invasive [4,8].

### 2.2. Study Cohorts

The clinicopathological and RNA-Seq data of the CRC, GC and PCa cohorts of the Cancer Genome Atlas (TCGA) and the Memorial Sloan Kettering Cancer Center (MSKCC) were utilised to interrogate the MPD phenomenon in primary or clinical cancer. All cancer cohort data utilised in this study were retrieved from the Genome Data Commons (GDC) and cBioPortal for Cancer Genomics databases [23,24,25,26,27,28]. 

### 2.3. Data Handling

Scripts and codes for data retrieval were written in Linux-based language in the Windows-based Ubuntu 20.04 environment. Gene expression datasets were prepared in accordance with the Molecular Signature Database [29,30] using Linux-based scripts. Excel spreadsheet was used to generate phenotype files, which were then converted to. cls files. The GC cohort comprised of 441 primary cases with clinicopathological and RNA-Seq data. The generated expression dataset for this cohort comprised of 415 cases with 20,531 mRNA expression records. The CRC cohort included 629 primary cases with clinicopathological and mRNA expression data, while the final gene expression dataset comprised 537 cases with 60,483 gene expression records. The TCGA Firehose PCa cohort comprised 500 primary cancers with clinicopathological and RNA-Seq data [26,27], while the MSKCC PCa cohort [28] had 150 cases comprising 131 primary and 19 metastatic tumours with matching clinicopathological and mRNA microarray expression profiling. The expression data underwent zero-to-one normalisation before utilisation in the downstream analysis. Furthermore, to enable appropriate comparisons, the generated gene expression signatures (see below) also underwent normalisation by fractional ranking per cohort.

### 2.4. Study Approach

The proliferation and migration phenotypes were measured by gene expression signatures, which were generated using the geometric mean of the expression values of the genes that subserve each phenotype. The CIN70 signature was used as the proliferation index in this study. This signature was originally designed as a surrogate for cell aneuploidy but has been found to represent proliferation indices in CRC and other cancers [31,32]. Gene ontology enrichment analyses [33] confirmed that the composite genes of the CIN70 signature generated are relevant to cell proliferation, cell replication, DNA metabolism and other biological processes involved in cell proliferation (See Appendix A). The migration/invasion index was generated using the expression values of the EMT regulators *SNAIL*, *SLUG*, *TWIST1*, *TWIST2*, *ZEB1*, *ZEB2*, *GLI1*, *GLI2* and *KLF4* [34,35,36]. Proliferation and migration indices were generated from the geometric means of the expression levels of the component genes of the indices as previously described [37,38]. Then, correlative analysis and the chi-square test of independence were used to determine the type of relationship, if any, existing between the proliferation and migration indices within cancer stages and between the early- and late-stage cancers in the cancer cohorts. The relationship type between migration and proliferation within the molecular subtypes of CRC and GC was also determined. Moreover, a third index—the dichotomy index (DI)—was derived for each case by calculating the ratio of the proliferation and migration indices. A high DI represented cases with high proliferation and low migration indices, while a low DI meant cases had low proliferation and high migration indices. The DI was used to determine the change in proliferation or migration dominance within and between tumour stages. Gene set enrichment analyses (GSEA) were performed using the cell proliferation and migration gene sets to confirm the differential enrichment of proliferation and migration between tumour stages. The significantly enriched core genes were subjected to gene ontology enrichment analysis to validate the differential enrichment of proliferation- and migration-based biological, molecular and cellular functions between cases with high and low DI.

### 2.5. Statistical Analysis

The chi-square (or Fisher) test in SPSS Version 29 was used to probe for significant associations between categorical variables, while bivariate correlative analysis was used to test the correlations between continuous variables. The independent *t*-test was used to measure the mean differences in continuous variables between discrete groups. Specifically, the independent T and chi-square tests were used to interrogate the presence of a shift from proliferation dominance to migration dominance across the tumour stages. A Z-score statistics calculator (https://www.socscistatistics.com/tests/ztest/default2.aspx, accessed on the 25 February 2024) was used to compare the ratios between the discrete categories. A *p* value of <0.05 was used as the threshold for significant tests, while the Benjamini-Hochberg correction was applied to correct for multiple testing at an FDR of 0.05. GSEA were performed as a phenotype permutation, using a nominal *p* value of 0.05 and an FDR of 0.25 as thresholds, according to the default software parameters.

## 3. Results

### 3.1. Demographics of Cancer Cohorts

All 626 CRC cases had complete data regarding the tumour, node and metastasis (TNM) status, while 533/629 cases had data on proliferation (CIN70), migration (EMT) and DI indices. The CRC cohort was categorised into early-(TNM Stages I and IIA, *n* = 333) and late-(TNM Stages IIB-IV, *n* = 293) stage diseases. In the GC cohort, 432/441 cases had complete TNM data, while 410/441 had proliferation, migration and DI signature data. The GC cohort was also classified into early-(Stage IA and IB, *n* = 59) and late-(Stage II-IV, *n* = 382) stage diseases. The TCGA PCa cohort was re-categorised into early (TNM I-IIIA, *n* = 196) and late (IIIB-IV, *n* = 260) cancer stages. Staging data were not available for 104 cases. The MSKCC primary cancer cohort had 85 early-stage and 46 late-stage cases, as well as 19 metastasis cases. Kaplan-Meier analyses confirmed that these re-categorisations of TNM stages have valid biological or clinical relevance (Figure 1) and can thus be applied to interrogating the migration–proliferation relationships in primary gastrointestinal cancers.

### 3.2. Migration–Proliferation Correlation in CRC

Interrogating the CRC cohort revealed an inverse correlation between the proliferation and migration indices within both TNM stages of CRC, as well as in the overall cohort. (Table 1 and Figure 2). An independent t-test confirmed that within and across tumour stages, low migration correlated significantly with high proliferation and vice versa (Figure 3: CRC cohort, upper panel). The independent T and chi-square tests also showed a possible correlation between DI and tumour stage, with a reduction in the DI from early- to late-stage diseases, which is evidence of a shift from proliferation dominance to migration dominance in late tumour stages (chi-square test, X^2^ = 2.735, *p* = 0.098; Figure 3: CRC cohort, lower panel). Nevertheless, this correlation did not achieve statistical significance (see Table 2). However, GSEA and gene ontology analysis showed that the CRC subset which had a high DI was significantly enriched for gene sets with biological processes, molecular functions, and cell components that subserve the cell proliferation programme (Appendix A), while the low-DI CRC subset showed enrichment for gene sets whose functions are associated with the cell invasion/migration programme (Appendix A). Furthermore, parallel to the high-DI subset, the early-stage CRC subset was enriched for the gene sets involved in cell proliferation (Appendix A). In contrast, the late-stage CRC subset was enriched in the migration/invasion gene sets, similar to the low-DI subset (Appendix A). To confirm that the DI status matches the disease stage, the core enrichment genes of each gene set were compiled for the four CRC categories (high- and low-DI status and early- and late-stage disease; see Appendix A). While 113/1721 genes were commonly enriched for the early-CRC and low-DI categories, 386/1490 enriched genes were shared between the early-CRC and high-DI categories (z-score = 15.086; *p* < 0.001). In contrast, 355/1404 genes were commonly enriched between the combined low-DI and late-stage disease categories, whereas 39/1762 shared genes were found in the high-DI and late-stage categories (z-score =19.538; *p* < 0.001). These results suggested that the reduction in cell proliferation observed in late-stage CRC was related to cell migration/invasion, which increased from early- to late-stage CRC. The gene set and ontology enrichment analyses results validated the results of the correlation and chi-square analyses. Overall, the results were in keeping with the MPD model in CR, but excluded both the SMP and PR models of migration–proliferation relationships in cancer cells. 

### 3.3. Migration–Proliferation Correlation in GC

Inverse correlations between proliferation and migration/invasion indices were observed within the TNM stages, as well as in the overall GC cohort (Table 1 and Figure 3: GC cohort, Figure 4). An independent t-test also confirmed an inverse relationship between migration and proliferation (Figure 3, GC cohort). An independent T-test of the DI pattern showed a statistically significant tendency to decrease from the early to late tumour stages in the GC cohort (Table 2). This pattern was confirmed by a chi-square test of the relationship between DI and the stages of cancer (chi-square test: X^2^ = 12.725; *p* < 0.001, Figure 3), indicating that the decrease in cell proliferation from early- to late-stage diseases is related to the increase in cell migration/invasion in the same direction, which is in keeping with the shift from proliferation dominance to migration dominance in late-stage cancer. Furthermore, the high-DI GC subset exhibited an enrichment of proliferation gene sets similar to the early-stage cancer subset, while the low-DI and late-stage disease subsets showed identical enrichment of migration/invasion gene sets (see Appendix A). The results support the MPD model in GC but not in the SMP and PR models.

### 3.4. Migration–Proliferation Correlation in PCa

Bivariate analyses revealed an inverse relationship between the proliferation and migration indices in both PCa cohorts, thereby validating MPD in non-gastrointestinal cancers. Furthermore, within the cancer stages, proliferation showed an inverse correlation with migration, which is consistent with the findings observed for GC and CRC (Table 1). In addition, both the early and late stages of PCa maintained an inverse proliferation-migration correlation, as before. Moreover, an inverse proliferation-migration correlation was demonstrated in primary and metastatic tumours in the MSKCC cohort. However, in contrast to the proliferation-to-migration dominance shifts observed for GC and CRC, no such shift was observed for any of the PCa cohorts, as no significant difference in DI scores was observed for either proliferation or migration between tumour stages in either PCa cohort (Table 3). GSEA demonstrated no differential enrichment of the biological or molecular processes of proliferation and migration between tumour stages in either PCa cohort. On the other hand, between primary and metastatic tumours in the MSKCC cohort, there was an enhancement of the proliferation dominance. More cases in the metastasis cohort had higher DI scores than the cases in the primary cohort (Table 3, Figure 5), in keeping with the higher proliferative state of cancer cells at metastatic sites.

### 3.5. Migration–Proliferation Correlation within Molecular Subtypes of Cancer

The TCGA CRC and GC cohorts were subtyped into chromosomal instability (CIN, CRC and GC), microsatellite instability (MSI, CRC and GC), genome-stable (GS, CRC and GC), POLE (CRC and GC) and Epstein-Barr (EBV, GC only) subtypes. A total of 446/629 cases in the CRC cohort had molecular subtype data, including 316/446 CIN, 64/446 MSI, 58/446 GS and 9/446 POLE CRC data. In the GC cohort, there were 223/383 CIN, 73/383 MSI, 50/383 GS, 30/383 EBV and 7/383 POLE subtypes. Correlation analyses showed an inverse relationship between proliferation and migration/invasion indices in the CIN (R = −0.383, *p* < 0.001) and MSI (R = −0.265, *p* = 0.034) subtypes of CRC. However, the GS (R = 0.112, *p* = 0.380) and POLE (R = −0.021, *p* = 0.958) subtypes showed no statistically significant relationship between proliferation and migration indices. In the GC cohort, inverse relationships between proliferation and migration/invasion indices were observed for all molecular subtypes except POLE (CIN: R = −0.525, *p* < 0.001; MSI: R = −0.370, *p* = 0.002; GS: R = −0.686, *p* < 0.001; EBV: R = −0.641, *p* < 0.001; POLE: R = −0.584, *p* = 0.187). In both the CRC and GC cohorts, the POLE molecular subtype exhibited an inverse migration−proliferation relationship, but not at statistically significant levels, probably because this subtype occurred in very few cases. The overall results demonstrate that MPD exists in CRC and GC, irrespective of the molecular subtypes of these cancer types.

## 4. Discussion 

The primary aim of this study was to confirm the MPD phenomenon in natural cancers, that is, in cancer growing in the natural settings of stromal cells or the tumour microenvironment. The use of in vitro techniques for this study would have completely defeated this aim. Whereas MPD has been validated several times in cancer cell lines, the phenomenon has not been studied in natural cancers [3,4,5,6,7,8,9,20]. In addition, the fact that stromal cells or the tumour microenvironment participate in the promotion of tumour progression [39,40,41,42] makes the use of bulk tumours more relevant for the study of the MPD phenomenon in primary cancers. The cross-talk between the stroma and the tumour parenchyma provides a cue for the proliferation-migration switch and the reverse migration–proliferation switch, as well as influences the cancer cell phenotype [3,8,9,10,11,43,44], just as tumour cells subvert the stroma and make it permissive and supportive for cancer cells [42]. Therefore, the “contaminating” stroma forms part of the MPD mechanism in bulk tumours. It has been established that the subsisting cancer cell phenotype is a product of parenchymal and stroma activities [42,44]. The measured indices used as surrogates for cell proliferation and migration/invasion in this study are consequences of the bulk of the tumour. Hence, the presence of the stroma cannot be regarded as a confounding factor. 

The MPD phenomenon has not been comprehensively demonstrated in primary (or clinical) cancers. Using gene expression signatures as surrogates for cell proliferation and migration/invasion in large GC, CRC, and PCa cohorts, this study demonstrated that an inverse relationship and a reverse association between migration and proliferation exist within tumour stages and between early- and late-stage diseases. It also demonstrated a shift from proliferation dominance to migration dominance in late-stage gastrointestinal cancer. Overall, the results were consistent with the MPD model rather than the SMP or PR models of cell proliferation and migration. This conclusion is valid regardless of the origin-of-metastasis models in which the results of the study are situated. For example, the clonal evolution model states that invasive or metastatic cell clones arise from primary tumours after acquiring sufficient genetic alterations that confer metastatic potentials [45]. If it is assumed that the metastatic clone has an SMP phenotype, then the inverse migration–proliferation correlation observed in the cancer cohorts in this study would be absent, especially in the progressive stages of cancer, although cell proliferation equalises cell migration at every turn. The shift from proliferation dominance to migration dominance observed in late-stage disease would also be non-existent for the same reason. Similarly, if the PR model is assumed within the clonal evolution theory, not only would an increase in migration and proliferation occur concurrently, albeit in different clones of cells; however, the turnover of proliferative cells (due to cell death from diminishing nutrients and oxygen) and migratory cells (due to net migration of cells out of the tumour body) would also proceed simultaneously. These two events would equilibrate the proliferative and migratory phenotypes, preclude the inverse migration–proliferation correlation, and shift from migration dominance to proliferation dominance observed in this study. Furthermore, the alternative origin-of-metastasis hypothesis [46], which states that subpopulations of cancer cells have a predilection for metastasis ab initio and that this predilection manifests following environmental cues, concurs with the MPD (and SMP) model because it also posits that indolent metastatic cells possess the ability for outgrowth or proliferation at the metastatic site and in response to microenvironmental stimuli emanating from the primary tumour [46]. However, the alternative hypothesis excludes the PR model. In addition, the background genetic variation hypothesis [47], which states that germline variations in patients with cancer result in variations in gene expression in transformed cells, in which gene expression variations confer metastatic potential on transformed cells, presupposes that all cancer cells within a tumour body possess an innate ability to metastasise, in keeping with the MPD and SMP models. Finally, the stem cell hypothesis states that a body of cancer arises from a hierarchy of cells that possess metastatic and proliferative potential [48], in agreement with both the MPD and SMP models. However, for the latter origin-of-metastasis hypotheses, the CRC and GC data in this study support MPD over SMP.

The only scenario in which an inverse or indirect migration–proliferation relationship would exist within each tumour stage at the cell population level (and a shift from proliferation-to-migration dominance in late-stage tumours) would be if the cells exhibit MPD at their individual levels. This would be the case because if the predominance of one phenotype diminishes the other (as the MPD principle implies) [3,5], then both traits are constantly in an inverse relationship, irrespective of the tumour stage or subtype. In the SMP and PR models, the cells showed a positive or direct, rather than an inverse or indirect, correlation between migration and proliferation at the population level, irrespective of the tumour stage. Under no circumstances, the proliferation and migration phenotypes exhibit an indirect or inverse/reverse relationship if a tumour couples proliferation and migration at the cellular level [19,20] or if separate clones of cells exhibit proliferation or migration phenotypes at equal rates.

However, as noted by Kuznetsov and Kolobov [4], the proliferation-to-migration dominance shift may not be a sine qua non-feature of the MPD, as tumour progression in the setting of the MPD has been noted to proceed or evolve via three mechanisms, including maximisation of cell migration (leading to the proliferation-to-migration dominance shift in late-stage tumour), maximisation of proliferation rate, and non-dominant state of either proliferation or migration rates [4]. While the first mechanism was observed in the GC and CRC cohorts, the third mechanism was observed in the PCa cohorts. The preferred mechanism of tumour progression depends on other critical intra- and extra-tumoural factors, an important one being the topology of the tissue in which the cancer cells are growing [49]. Tissue topology exerts different mechanical stimuli on cancer cells within a tumour [49]. In vitro studies have demonstrated the differential preferences of cancer cells for either proliferation or migration based on the architectural properties of tissues, including their geometry, confinement, and fluidity [49,50,51,52]. It is conceivable that the complex branching and arborising glandular architecture of a solid organ such as the prostate may confer differential spatial dynamics and mechanical stimuli on the tumour cells than the simpler tubular gastric and colorectal tissues.

The present study data did not support the hypothesis that MPD may be absent in the mesenchymal-like subsets of GC and CRC, as the tendency for inverse migration–proliferation relationships was observed in all molecular subtypes of both cancers. 

However, not all studies in the scientific literature support the occurrence of the MPD phenomenon in cancer. While some melanoma cell line studies have demonstrated that malignant melanocytes display MPD [2], other studies have shown that these cancer cells exhibit simultaneous proliferation and migration [9,20]. Hence, further studies are needed to clarify the migration–proliferation relationship, at least in primary malignant melanoma.

The presence of MPD in cancer has implications for therapy, as it has been demonstrated that the treatment of cancer cells with common cytotoxic agents induces cell migration and invasion [53]. This implies that for more effective cancer cell killing, proliferation and migration/invasion must be differentially targeted by chemotherapy using regimens that target both proliferation and migration either simultaneously or sequentially [54].

## 5. Conclusions

This study has demonstrated that an inverse relationship exists between proliferation and migration indices in GC and CRC cohorts, and this relationship is observed within the different stages of cancer as well as between early- and late-stage cancers. These findings are consistent with the MPD phenomenon described in cancer cells in culture studies. They may also have implications for cancer therapy.

## Figures and Tables

**Figure 1 biology-13-00753-f001:**
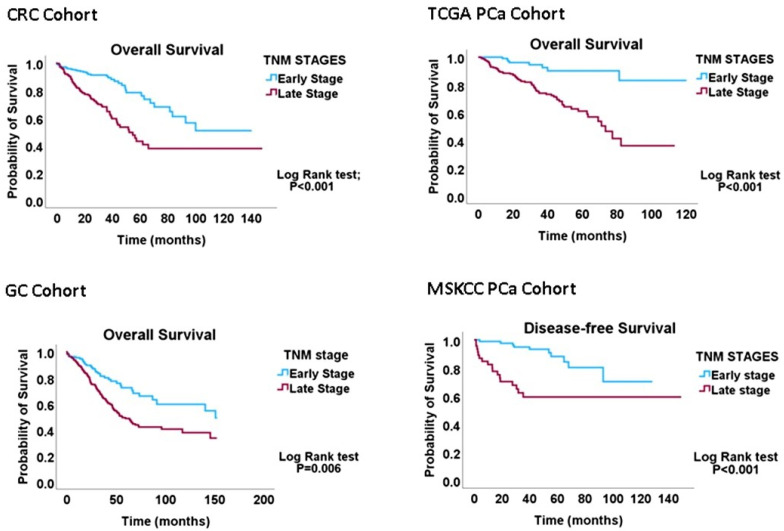
Kaplan-Meier graphs demonstrating that the early-versus-late cancer-staging scheme predicts overall survival in CRC, GC and PCa cohorts. The results validate the conversion of the four-tier staging scheme to the two-tier scheme utilised to probe the MPD phenomenon in this study.

**Figure 2 biology-13-00753-f002:**
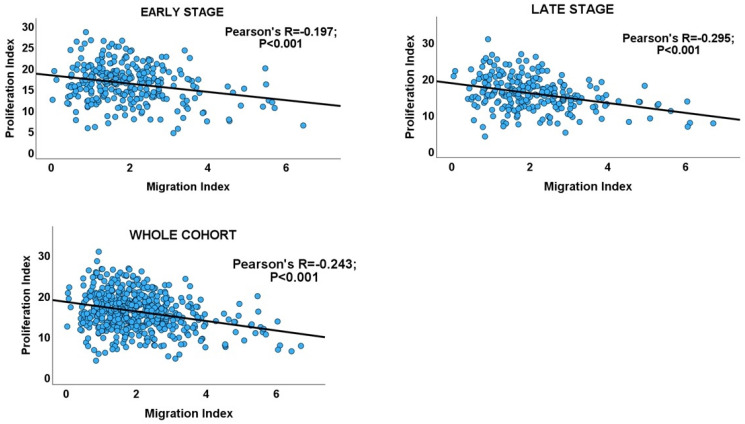
Scatterplots showing the inverse migration–proliferation correlation within the early and late stages of the CRC cohort and in the entire cohort.

**Figure 3 biology-13-00753-f003:**
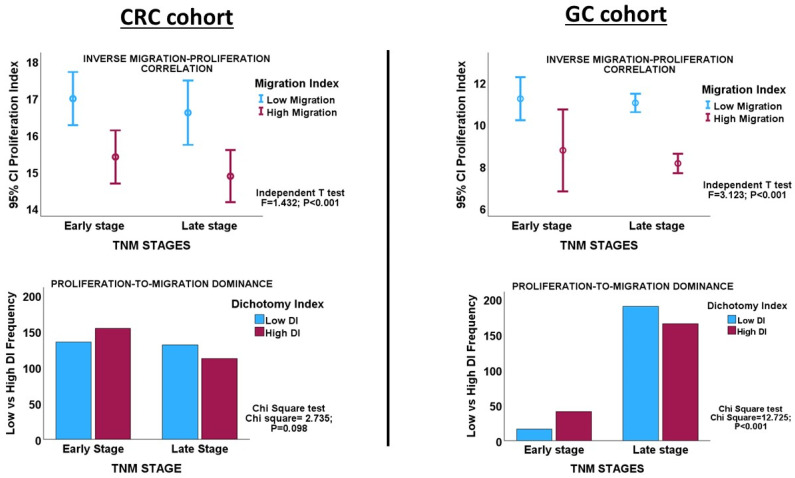
**Upper panels**. Error charts showing the statistically significant correlation between migration and proliferation, as demonstrated by the Independent *T*-test in the CRC and GC cohorts, respectively. CRC and GC cases with high proliferation indices have low migration indices, and vice versa. **Lower panels**. Clustered bar charts showing the tendency of tumour progression to shift from a proliferation dominance in early-stage cancer to a migration dominance in late-stage disease in the CRC and GC cohorts. High DI = high proliferation index; Low DI = high migration index.

**Figure 4 biology-13-00753-f004:**
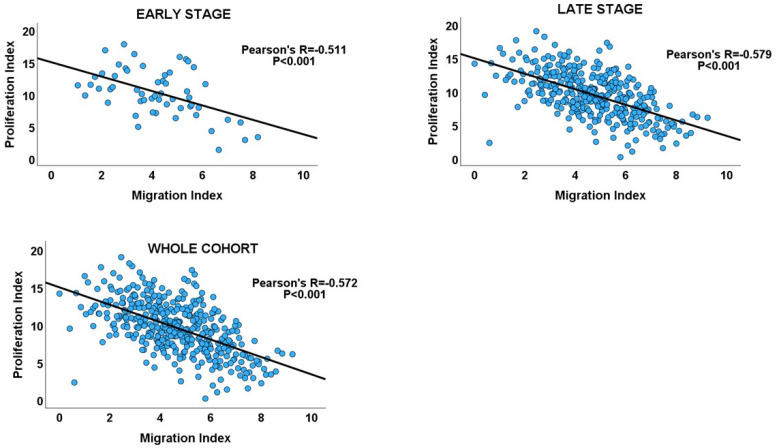
Scatterplots showing the inverse migration–proliferation correlation within the early and late stages of the GC cohort and in the entire cohort.

**Figure 5 biology-13-00753-f005:**
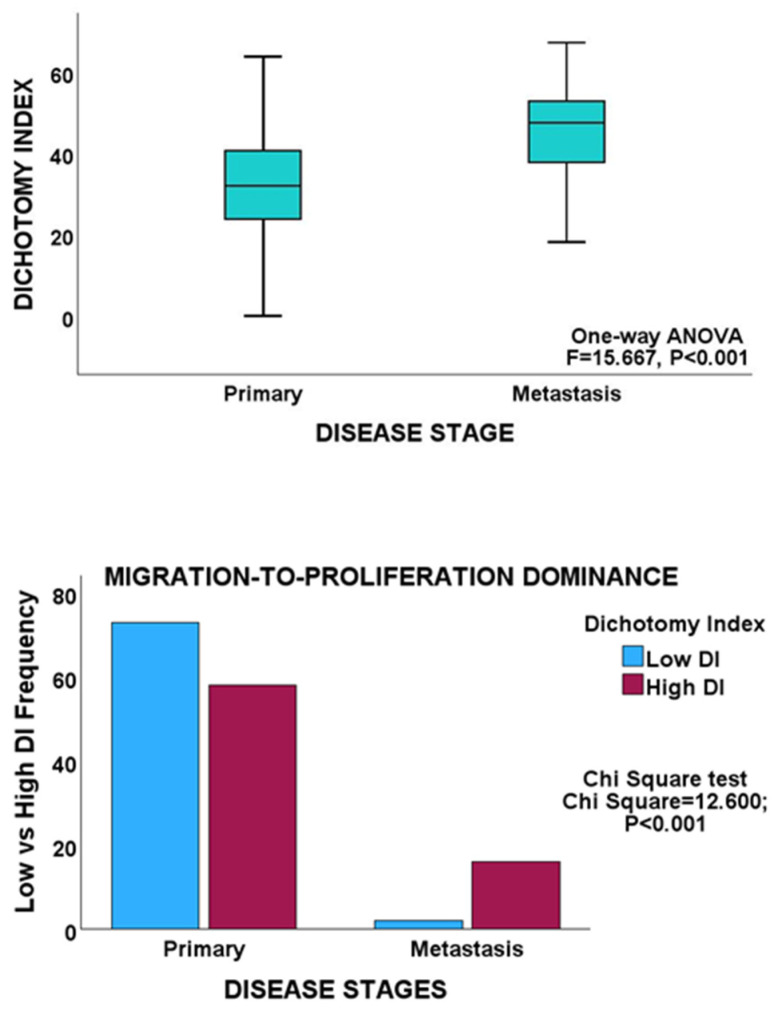
**Upper panel**, box plot showing a significantly higher mean DI in the metastasis group of the MSKCC cohort. **Lower panel**, clustered bar chart showing proliferation dominance in the metastasis cohort.

**Table 1 biology-13-00753-t001:** Stage-specific migration–proliferation correlation.

Cancer Stage	Number of Cases	Pearson’s Correlation (R)	95% Confidence Interval for R	*p* Value	Adjusted *p* Value
CRC cohort					
Early stage	290	−0.197	−0.305 to −0.083	<0.001	<0.001
Late stage	242	−0.295	−0.406 to −0.175	<0.001	<0.001
Overall	532	−0.243	−0.321 to −0.161	<0.001	-
GC cohort					
Early stage	55	−0.511	−0.681 to −0.280	<0.001	<0.001
Late stage	352	−0.579	−0.644 to −0.504	<0.001	<0.001
Overall	407	−0.572	−0.633 to −0.502	<0.001	-
TCGA PCa cohort					
Early stage	135	−0.396	−0.506 to −0.213	<0.001	<0.001
Late stage	258	−0.294	−0.402 to −0.178	<0.001	<0.001
Unclassified	104	−0.206	−0.384 to −0.013	<0.037	<0.037
Overall	497	−0.297	−0.358 to −0.96	<0.001	
MSKCC PCa cohort					
Early stage	85	−0.228	−0.420 to −0.015	0.036	0.036
Late stage	64	−0.407	−0.595 to −0.177	<0.001	0.002
Primary	131	−0.240	−0.396 to −0.070	0.006	0.010
Metastasis	18	−0.500	−0.784 to −0.043	0.013	0.016
Overall	149	−0.328	−0.465 to −0.176	<0.001	0.002

**Table 2 biology-13-00753-t002:** Differential Dichotomy Index between early and late-stage diseases in GC and CRC.

Cancer Stage	Number of Cases	Mean of DI	95% Confidence Interval for Mean DI	F	*p* Value
CRC cohort					
Early stage	289	26.206	18.645–33.767	3.418	0.065
Late stage	243	15.745	7.574–23.916		
Total	532	21.428	15.878–26.977		
GC cohort					
Early stage	17	14.233	10.852–17.615	23.682	<0.001
Late stage	387	6.379	5.728−7.029		
Total	404	6.709	6.055–7.634		

**Table 3 biology-13-00753-t003:** Differential Dichotomy Index in the PCa cohorts.

Cancer Stage	Number of Cases	Mean of DI	95% Confidence Interval for Mean DI	F	*p* Value
TCGA cohort					
Early stage	135	2.416	0.693–4.139	1.971	0.161
Late stage	258	4.390	2.600–6.180		
Total	393	3.712	2.398–5.026		
MSKCC cohort					
Early stage	85	2.073	1.098–3.048	2.620	0.108
Late stage	65	6.204	1.282–11.128		
Total	150	3.864	1.667–6.060		
Primary	131	2.343	0.175–4.511	15.667	<0.001
Metastasis	19	14.932	8.232–21.633		
Total	150	3.864	1.716–6.011		

## Data Availability

Publicly available datasets were analysed in this study. The datasets were freely available and can be accessed at https://portal.gdc.cancer.gov (accessed 25 February 2024) and https://www.cbioportal.org/ (accessed 25 February 2024).

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
