# Peer review of "Cell Migration–Proliferation Dichotomy in Cancer: Biological Fact or Experimental Artefact?"

_biology, 2024, doi:10.3390/biology13100753_

Round 1
Reviewer 1 Report (New Reviewer)
Comments and Suggestions for Authors The article presents an intriguing perspective on the proliferation, migration, and invasion of cancer cells, a crucial topic in the search for treatment strategies against this pathology. The authors offer a compelling non-experimental viewpoint that, if utilized effectively, can support various hypotheses. Overall, the study is innovative and interesting, utilizing a substantial amount of data. However, I believe it is not presented in the most optimal way, particularly regarding the use of tables and figures. The presentation of multiple tables and figures together makes it difficult to follow the thread of ideas. The figures vary in size, with some being very large and occupying space that could be better utilized. It would be beneficial not only to mention in the conclusion that “the study may have implications for cancer therapy” but also to discuss how this work can be applied to this critical aspect.Specific Comments: - The graphs in Figure 1 should be homogeneous; some (left side) appear blurry. - In line 237, there is an extra period before the parenthesis. - It is highly recommended that the figures have titles that highlight their significance. For example, Figure 3 begins with “Upper panels”; this presentation can be significantly improved. - Lines 303 to 319 are not justified, and there is a lot of text highlighted in yellow, making it look like a draft rather than the final version. - Figure 5 is poorly centered and is much larger compared to the others. - The references section is not justified. - The principles should not be included in the materials and methods section; they can be placed in the introductory part as a preamble to the study.

Author Response
Author's Reply to the Review Report (Reviewer 1)
Comment 1: The article presents an intriguing perspective on the proliferation, migration, and invasion of cancer cells, a crucial topic in the search for treatment strategies against this pathology. The authors offer a compelling non-experimental viewpoint that, if utilized effectively, can support various hypotheses. Overall, the study is innovative and interesting, utilizing a substantial amount of data. However, I believe it is not presented in the most optimal way, particularly regarding the use of tables and figures. The presentation of multiple tables and figures together makes it difficult to follow the thread of ideas.
The figures vary in size, with some being very large and occupying space that could be better utilized.
Response 1: The placement of the tables and figures in the manuscript were done in accordance with the template provided by the MDPI Biology journal. The presentation of multiple figures was done to demonstrate the multiple results/ideas expressed in a few Result subsections. The figures include simple statistical charts that are clearly and adequately annotated; they are easy to understand. The figure legends are standard. The tables are also adequately annotated. But I will defer to the journal editors if they see a need to alter the sizes of the figures or re-arrange the tables.
Comment 2: It would be beneficial not only to mention in the conclusion that “the study may have implications for cancer therapy” but also to discuss how this work can be applied to this critical aspect.
Response 2: The implications of the MPD with respect to cancer therapy was discussed in the Discussion section: “The existence of MPD in cancer has implications for therapy, as it has been demonstrated that the treatment of cancer cells with the common cytotoxic agents induces cell migration and invasion [54]. This implies that for more effective cancer cell killing, proliferation and migration/invasion must be differentially targeted by chemotherapy, using regimens that target both proliferation and migration either simultaneously or sequentially [54].” (lines 481-486)
Specific Comments:
Comment 3: - The graphs in Figure 1 should be homogeneous; some (left side) appear blurry.
Response 3: The Figures are all clear. No blurry figures in the manuscript submitted to the journal.
Comment 4: - In line 237, there is an extra period before the parenthesis.
Response 4: Corrected
Comment 5: - It is highly recommended that the figures have titles that highlight their significance. For example, Figure 3 begins with “Upper panels”; this presentation can be significantly improved.
Response 5: The Figure legends used in the manuscript follow the standard formats for legends. However, I have added further explanations of Figure 3 (lines 277-282) (highlighted in yellow)
Comment 6: - Lines 303 to 319 are not justified, and there is a lot of text highlighted in yellow, making it look like a draft rather than the final version.
Response 6: The manuscript was written in the journal’s template format. The lines and paragraphs have now been amended to follow the MDPI Biology format. The version of the manuscript reviewed by this reviewer is a re-submission. The highlighted text denoted the additions that were made to the older version of the manuscript prior to re-submission.
Comment 7: - Figure 5 is poorly centered and is much larger compared to the others.
Response 7: Figure 5 has now been appropriately centred following the template style of the MPDI Biology journal
Comment 8: - The references section is not justified.
Response 8: The reference section has been amended to follow the journal format
Comment 9: - The principles should not be included in the materials and methods section; they can be placed in the introductory part as a preamble to the study. peer-review-38844949.v1.pdf
Response 8: I believe that the Introduction, as it currently is, adequately lays the ground for the other sections of the manuscript. The "Principles" are the notions generated by the study to enable interrogation and clarifications of the subject-matter. These notions are based on scientific evidences which are referenced in the "Principles" subsection. I believe they should remain in the Materials and Methods section as they are rightly "Methods".
Reviewer 2 Report (New Reviewer)
Comments and Suggestions for Authors
This manuscript details the patterns of proliferation and migration in colorectal, gastric, and prostate cancer. While the manuscript is well-written, I have the following comments:
Typographical error in line 119
Typographical error in line 132
How does demographic and environmental stochasticity affect MPD?
What were the inclusion and exclusion criteria for cases in the cohorts?
References do not adhere to the journal format.
In the discussion section, please explain why proliferation-to-migration dominance shifts were not observed in the PCa cohort.
Author Response
Author's Reply to the Review Report (Reviewer 2)
This manuscript details the patterns of proliferation and migration in colorectal, gastric, and prostate cancer. While the manuscript is well-written, I have the following comments:
Comment 1: Typographical error in line 119.
Response 1: Corrected
Comment 2: Typographical error in line 132.
Response 2: Corrected
Comment 3: How does demographic and environmental stochasticity affect MPD?
Response 3: I do not know for a certain, but I have speculated, based on findings from previous studies, that the complexity of tissue topology determines the differential cellular phenotypes, including proliferation and migration. The effect of differential tissue topology on cancer phenotypes deserves further exploration.
Comment 4: What were the inclusion and exclusion criteria for cases in the cohorts?
Response 4: The inclusion criteria include the availability of most clinico-genomic data. The exclusion criteria were the absence of these data. All cases in all 3 cohorts were included, but cases with any missing data was excluded from the specific analyses by the SPSS data, especially in the TCGA CRC cohorts.
Comment 5: References do not adhere to the journal format.
Response 5: I have re-formatted the references to adhere to the journal’s referencing style
Comment 6: In the discussion section, please explain why proliferation-to-migration dominance shifts were not observed in the PCa cohort.
Response 6: According to Kuznetsov and Kolobov [4], tumour progression occurs by 3 mechanisms: maximization of cell migration (leading to the proliferation-to-migration dominance shift in late-stage tumour), maximization of proliferation rate, and non-dominant state of either proliferation or migration rates [23]. In this study, I found that both migration and proliferation (as judged by the expression signatures) increased from early to late disease in prostate cancer, and that the proliferation dominance was maintained from early to late disease stages. I went on to speculate that the absence of the proliferation-to-migration dominance shifts in the prostate cancer cohort could be due to the difference in tissue topology between stomach/colorectum (hollow organs) and the prostate (solid with arborizing glandular architecture). Hence, I discussed tissue topology as a determinant of cancer phenotype, and suggested that the differential topology between the GI organs and the prostate could explain the absence of the proliferation-to-migration dominance shifts in prostate cancer. These points were highlighted in the Discussion section.
Reviewer 3 Report (New Reviewer)
Comments and Suggestions for Authors
In general, the manuscript “Cell migration-proliferation dichotomy in cancer: biological fact or experimental artefact?” is well written and overall provides a good understanding and explanation of the problem.
The introduction part I think provides enough justification for exploring the migration proliferation dichotomy (MPD) on primary cancer cells, however I would expect to have a rationale on the reason to focus on GI and prostate cancers as study models.
Additionally, I think we should not generalize the findings of this study and apply the same conclusions into other cancer types if these have not yet been analyzed into the same parameters. As stated in the last paragraph on the introduction “The study hypotheses 81 are that the cell proliferation and migration programmes in individual cancer cells are 82 mutually exclusive to each other and that this mutual exclusivity is reflected as an in-83 verse, reverse or indirect correlation between proliferation and migration signatures at 84 the cell-population level in primary cancers”.
Methods are well explained. In the results section, specifically in figure one, the legend of each graph has two extra icons with no explanation of their meaning.
Including in-vitro samples of the same tumor types as controls I think will provide an interesting comparative on how much of a predictive parameter can this type of samples be for assessing MPD, or how much similarity or disparity we find between samples of the same cancer type but arising from different settings (cell line vs clinical tumor).
Overall, the discussions and conclusions are consistent with the evidence and arguments presented in the manuscript. I think the paragraph added about the implications of this study is especially important and highlights the importance of applying this tool to a broader variety of cancers.
My Overall recommendation is to Accept after Minor Revisions, specifically enriching the introduction section.
Author Response
Author's Reply to the Review Report (Reviewer 3)
Comment 1: In general, the manuscript “Cell migration-proliferation dichotomy in cancer: biological fact or experimental artefact?” is well written and overall provides a good understanding and explanation of the problem.
Response 1: Thanks for the positive review
Comment 2: The introduction part I think provides enough justification for exploring the migration proliferation dichotomy (MPD) on primary cancer cells, however I would expect to have a rationale on the reason to focus on GI and prostate cancers as study models.
Response 2: The reason for choosing gastric, colorectal and prostate cancers is that they are my focus of research. Therefore, the cancer types were chosen randomly. However, with the findings from this study, I now have a basis for interrogating other cancer types for the existence of the MPD phenomenon.
Comment 3: Additionally, I think we should not generalize the findings of this study and apply the same conclusions into other cancer types if these have not yet been analyzed into the same parameters. As stated in the last paragraph on the introduction “The study hypotheses 81 are that the cell proliferation and migration programmes in individual cancer cells are 82 mutually exclusive to each other and that this mutual exclusivity is reflected as an in-83 verse, reverse or indirect correlation between proliferation and migration signatures at 84 the cell-population level in primary cancers”.
Response 3: The study only interrogated gastric, colorectal and prostate (primary and metastatic) cancers. Hence, no generalization was made or extended to other cancers at the end of the study. At the tail of the Discussion section, I stated that “however, not all studies in the scientific literature support the occurrence of the MPD phenomenon in cancer. While some melanoma cell line studies have demonstrated that malignant melanocytes display MPD [2], other studies have shown that these cancer cells exhibit simultaneous proliferation and migration [9, 20]. Hence, further studies are needed to clarify the migration–proliferation relationship, at least in primary malignant melanoma.” The statement of hypothesis did not seek to generalise the findings of the study.
Comment 4: Methods are well explained. In the results section, specifically in figure one, the legend of each graph has two extra icons with no explanation of their meaning.
Response 4: The icons have no place in the survival charts and have been removed from the Figure
Comment 5: Including in-vitro samples of the same tumor types as controls I think will provide an interesting comparative on how much of a predictive parameter can this type of samples be for assessing MPD, or how much similarity or disparity we find between samples of the same cancer type but arising from different settings (cell line vs clinical tumor).
Response 5: The aim of the study is to investigate the MPD in clinical or natural cancer using bioinformatics methodology. The MPD has been severally interrogated in cell lines of glioblastoma, melanoma, prostate cancer, breast cancer, etc. So, this study wanted to see if that phenomenon exists in clinical cancer, since findings from in vitro studies may not accurately represent the biology of natural cancers for some reasons stated in the manuscript. The use of in vitro techniques would have defeated the aim of this study. The CIN70 and EMT gene expression signatures were used as surrogates for proliferation and migration, respectively. These signatures have also been confirmed to represent cell proliferation and cell migration by many studies.
Comment 6: Overall, the discussions and conclusions are consistent with the evidence and arguments presented in the manuscript. I think the paragraph added about the implications of this study is especially important and highlights the importance of applying this tool to a broader variety of cancers.
Response 6: Thank you.
Recommendation: My Overall recommendation is to Accept after Minor Revisions, specifically enriching the introduction section.
Round 2
Reviewer 1 Report (New Reviewer)
Comments and Suggestions for Authors
The work submitted by the researchers, together with their responses to the comments, meets the requirements for the publication of the article in question.
The data presented, as well as the format, are appropriate and I believe they provide valuable information on the topic analyzed. The necessary specific comments have been addressed, and additional comments have been made for better understanding.
The only extra comment that needs to be reviewed is that figure 1 seems to be poorly placed, since it is in the middle of the figure and its title one line of text (line 212).
This manuscript is a resubmission of an earlier submission. The following is a list of the peer review reports and author responses from that submission.
Round 1
Reviewer 1 Report
Comments and Suggestions for Authors
The paper studies the migration-proliferation dichotomy in primary and clinical cancer by calculating the migration/invasion indexes and analyzing their relationship within tumour stages and between early- and late-stage diseases. The paper is interesting but requires improvement.
Specific Comments:
1) An example of how migration/invasion indexes are generated from datasets must be provided.
2) The supplementary files (Supplementary Materials_CIN70_GOEA, Supplementary Materials_GSEA_CRC_DI_HIGH, Supplementary Materials_GSEA_CRC_DI_LOW, Supplementary Materials_GSEA_CRC_EARLY, Supplementary Materials_GSEA_CRC_LATE, Supplementary Materials_ENRICHED_GENE_LISTS, and SHARED_GENE_LISTS) are missing. Please add information on the contents of each row in these files and provide instructions on how to use them.
3) Lines 174 to 188 must be deleted. The paper needs thorough proofreading.
4) It is unclear what frequency is plotted in the lower panels of Figure 3. These panels appear to support the claim of a shift from proliferation dominance in early-stage cancer to migration dominance in late-stage disease for gastric cancer (GC) but not for colorectal cancer (CRC) cohorts. These charts should include error bars to improve data clarity and reliability.
Author Response
The paper studies the migration-proliferation dichotomy in primary and clinical cancer by calculating the migration/invasion indexes and analyzing their relationship within tumour stages and between early- and late-stage diseases. The paper is interesting but requires improvement.
Specific Comments:
1) An example of how migration/invasion indexes are generated from datasets must be provided.
Response 1:
The proliferation and migration indices were generated from the geometric means of the expression levels of the component genes of the indices. A statement of the method of signature generation, with references, has been added to the revised manuscript. Also, Supplementary Materials_CIN70_GOEA has been added to the supplementary data.
2) The supplementary files (Supplementary Materials_CIN70_GOEA, Supplementary Materials_GSEA_CRC_DI_HIGH, Supplementary Materials_GSEA_CRC_DI_LOW, Supplementary Materials_GSEA_CRC_EARLY, Supplementary Materials_GSEA_CRC_LATE, Supplementary Materials_ENRICHED_GENE_LISTS, and SHARED_GENE_LISTS) are missing. Please add information on the contents of each row in these files and provide instructions on how to use them.
Response 2:
All the supplementary data was submitted as Excel spreadsheet with the manuscript. The complete supplementary data has been re-submitted with the revised manuscript.
3) Lines 174 to 188 must be deleted. The paper needs thorough proofreading.
Response 3:
Deletion done. The manuscript has been sent for proofreading. The certificate of proofreading has been submitted with the revised manuscript.
4) It is unclear what frequency is plotted in the lower panels of Figure 3. These panels appear to support the claim of a shift from proliferation dominance in early-stage cancer to migration dominance in late-stage disease for gastric cancer (GC) but not for colorectal cancer (CRC) cohorts. These charts should include error bars to improve data clarity and reliability.
Response 4:
The “Low vs High DI Frequency” in the Y-axis of the Figure 3 lower panel represents the categorized Dichotomy Index (or DI) count. As was explained in the Method section of the manuscript, the DI is a ratio of the CIN70 and the EMT indices. It was generated to enable the interrogation of the shift from proliferation- to migration-dominance. The conversion of the DI to a categorical variable enabled easy comparison of migration and proliferation, by chi square test, between disease stages.
The chi square results showed that in the CRC cohort, there was only a tendency for tumour progression to shift from proliferation- to migration-dominance because the shift did not achieve statistical significance. However, early-stage CRC was significantly enriched for genes that subserve cell proliferation, while late-stage CRC showed significant enrichment of genes with migration function. Further analysis of the GSEA results confirmed the proliferation- to migration-dominance shift from early- to late-stage disease.
The Figure 3 lower panel is a clustered bar chart derived from comparing two categorical variables using chi square test. Hence the bar charts do not carry error bars.
Reviewer 2 Report
Comments and Suggestions for Authors
The author use the public data sets from TCGA to compare the relation of proliferation and migration. Author fund that proliferating cancer cells maintain low level of migration ability. However, there is lack of in vitro study or clinical study to prove that. The bioinformatic analysis is not enough to conclude this hypothesis. Also, there is missing explanation of diving the data sets into two group, early stage and late stage is lacking of explanation. Why is the cutoff set between IIA and IIB? Bulk RNAseq results are not good data set for this hypothesis. Single cell RNAseq data should be used to do the analysis.
Comments on the Quality of English Languageneed to be improve
Author Response
The author use the public data sets from TCGA to compare the relation of proliferation and migration. Author fund that proliferating cancer cells maintain low level of migration ability. However, there is lack of in vitro study or clinical study to prove that. The bioinformatic analysis is not enough to conclude this hypothesis. Also, there is missing explanation of diving the data sets into two group, early stage and late stage is lacking of explanation. Why is the cutoff set between IIA and IIB? Bulk RNAseq results are not good data set for this hypothesis. Single cell RNAseq data should be used to do the analysis.
Response 1:
The primary aim of the study is to confirm the MPD phenomenon in natural cancers, i.e., in cancer growing in the natural settings of the stromal cells or tumour microenvironment. The use of in vitro techniques for this study would have completely defeated this aim. Whereas MPD has been validated several times in cancer cell lines, the phenomenon has not been studied in natural cancers (3-9). Plus, the fact of stromal cells or the tumour microenvironment participating in the promotion of tumour progression [40-43]. makes the use of bulk tumour more relevant for the study of the MPD phenomenon in primary cancers. The cross-talk between the stroma and the tumour parenchyma provides the cue for the proliferation-migration switch and the reverse migration-proliferation switch and influence the cancer cell phenotype [3, 8-11, 44, 45], just as tumour cells subvert the stroma and make it permissive and supportive for the cancer cells [43]. Therefore, the “contaminating” stroma form part of the MPD mechanism. It has been established that the subsisting cancer cell phenotype is a product of parenchymal and stroma activities [43, 45]. The measured indices used to represent cell proliferation and migration/invasion in this study are consequences of the bulk of the tumour. Hence, the presence of the stroma cannot be regarded as a cofounding factor.
If an observed phenomenon at the single cell level cannot be replicated at the bulk cell population level, chances are that such a phenomenon is an artefact of cell culture experiments. Furthermore, the fact that the study, which used a combination of RNASeq and clinical data, confirmed the MPD phenomenon in cancer cells growing among “normal” stromal cells, and corroborated the results of many cell culture studies, shows that bioinformatics analysis of gene expression signatures from bulk tumours is an authentic approach to confirm, validate or corroborate single cell RNASeq studies.
The TCGA cancer cohorts have comprehensive genomic and clinicopathological data, hence researches carried out with these data are bona fide clinical studies.
The IIB stage of CRC represent the stage when “the cancer has grown through the wall of the colon or rectum” (https://www.cancer.org/cancer/types/colon-rectal-cancer/detection-diagnosis-staging/staged.html), hence it is at the point when tumour cells have breached the bowel wall and are no longer confined to the bowel. This stage was used as the cut-off for early- versus late-stage CRC in this study, and the dichotomy was validated with follow-up data (Figure 1).
Comments on the Quality of English Language
need to be improve
Response:
The manuscript has been subjected to further proofreading to improve the English language used. The certificate of proofreading has been submitted with this revised manuscript.
Reviewer 3 Report
Comments and Suggestions for Authors
The authors have raised an interesting point on the migration-proliferation dichotomy (MPD) in cancer. However, several important aspects need to be addressed and discussed:
1, while gene expression index/signatures were used to evaluate cell proliferation and migration from bulk patient-derived RNA-seq TCGA datasets, the purity of the tumor content needs to be considered. The presence of mixed “normal” tissues can act as significant confounding factors.
2, the study focuses on GC and CRC, which are primarily carcinomas. It would be valuable to discuss whether these observations can be extended to other types of carcinomas, such as prostate cancer.
3, the Cancer Cell Line Encyclopedia (CCLE) also has expression data available— could be beneficial to explore whether the findings can be applied to GC and CRC-lineage cancer cell lines.
4, the authors should consider applying the index to metastatic-related features.
Author Response
The authors have raised an interesting point on the migration-proliferation dichotomy (MPD) in cancer. However, several important aspects need to be addressed and discussed:
1, while gene expression index/signatures were used to evaluate cell proliferation and migration from bulk patient-derived RNA-seq TCGA datasets, the purity of the tumor content needs to be considered. The presence of mixed “normal” tissues can act as significant confounding factors.
Response 1:
The primary aim of the study is to confirm the MPD phenomenon in natural cancers, i.e., to validate the phenomenon in cancer cells growing in the natural settings of the stromal microenvironment. The use of in vitro single cell RNASeq approach would have completely defeated this aim. Plus, the tumour microenvironment (also known as “normal” stroma) is now known to be an active participant in the promotion of tumourigenesis and tumour progression and to influence the phenotype of the tumour parenchyma [40-45]. Hence, the presence of “impure”, “normal” stromal tissues within the bulk tumour cannot be regarded as confounding factors. Based on the knowledge of the participation of the tumour stroma in tumour promotion, any research work worth its salt should incorporate the contribution of the stromal cells to any tumour-progression phenomenon under study.
2, the study focuses on GC and CRC, which are primarily carcinomas. It would be valuable to discuss whether these observations can be extended to other types of carcinomas, such as prostate cancer.
Response 2:
Although the study used primary CRC and GC data to demonstrate the MPD phenomenon, it is assumed that the MPD phenomenon is also operative in other cancers such as prostate cancer. Hence the discussion of the controversy around the in vitro studies of the MPD in malignant melanoma. The current study has thus provided the basis for subsequent investigation of the MPD in other primary carcinomas.
3, the Cancer Cell Line Encyclopedia (CCLE) also has expression data available— could be beneficial to explore whether the findings can be applied to GC and CRC-lineage cancer cell lines.
Response 3:
The MPD phenomenon has been severally studied and demonstrated in cancer cell lines under simulated conditions of proliferation and invasion. The aim of this current study is to corroborate the phenomenon in cancer growing in the natural settings, i.e., in patients. The gene expression indices used to represent cell proliferation and invasion/metastasis in this study have been severally confirmed in cancer cell lines.
4, the authors should consider applying the index to metastatic-related features.
Response 4:
Two indices were utilized in this study to demonstrate MPD phenomenon in natural cancers: the CIN70 (proliferation) and the epithelial-mesenchymal transition (EMT) indices. The EMT index has long been established to represent invasion/metastasis. Tumour progression, in the form of tumour stage advancement was used as a surrogate for metastatic feature in the primary cancers in this study.
Round 2
Reviewer 2 Report
Comments and Suggestions for Authors
Accepted.
Reviewer 3 Report
Comments and Suggestions for Authors
The authors failed to address the concerns raised.